# A multi-pollutant and multi-sectorial approach to screen the consistency of emission inventories

Philippe Thunis[1] Alain Clappier[2], Enrico Pisoni[1], Bertrand Bessagnet[1] Jeroen Kuenen[3], Marc
Guevara[4], Susana Lopez-Aparicio[5]

[1] European Commission, Joint Research Centre, Ispra, Italy
[2] Université de Strasbourg, Laboratoire Image Ville Environnement, Strasbourg, France
[3] TNO, Department of Air, Climate and Sustainability, Utrecht, The Netherlands
[4] Barcelona Supercomputing Center, Barcelona 08034, Spain
[5] NILU – Norwegian Institute for Air Research, 2027 Kjeller, Norway

*Correspondence to*: Philippe Thunis (philippe.thunis@ec.europa.eu)

**Abstract:** Some studies show that significant uncertainties affect emission inventories, which may impeach conclusions based on air quality model results. These uncertainties result from the need to compile a wide variety of information to estimate an emission inventory. In this work, we propose and discus a screening method to compare two emission inventories, with the overall goal of improving the quality of emission inventories by feeding back the results of the screening to inventory compilers who can check the inconsistencies found and where applicable resolve errors. The method targets three different aspects: 1) the total emissions assigned to a series of large geographical area, countries in our application; 2) the way these country total emissions are shared in terms of sector of activity and 3) the way inventories spatially distribute emissions from countries to smaller areas, cities in our application. The first step of the screening approach consists in sorting the data and keep only emission contributions that are relevant enough. In a second step, the method identifies, among those significant differences, the most important ones that are evidence of methodological divergence and/or errors that can be found and resolved in at least one of the inventories. The approach has been used to compare two versions of the CAMS-REG European scale inventory over 150 cities in Europe for selected activity sectors. Among the 4500 screened pollutant-sectors, about 450 were kept as relevant among which 46 showed inconsistencies. The analysis indicated that these inconsistencies were almost equally arising from large scale reporting and spatial distribution differences. They mostly affect $SO_2$ and PM coarse emissions from the industrial and residential sectors. The screening approach is general and can be used for other types of applications related to emission inventories.

**Keywords**: emission inventories, quality assurance, screening, urban emissions

## 1. Introduction

Air pollution remains a critical issue, as it is one of the main causes of human health damage worldwide. In the EU28 alone, exposure to a pollutant such as fine particulate matter ($PM_{25}$) is estimated to be responsible for approximately 390 000 premature deaths per year (EEA, 2020). Reducing pollution levels requires appropriate regulatory decisions leading to the implementation of effective abatement strategies. Such decisions are not easy to support because they may involve several pollution sources interacting through complex and non-linear atmospheric phenomena. As only air quality models can simulate the impacts of emission reductions considering the complexity of the atmosphere, they are potentially the only tools able to support the planning of reduction strategies. The accuracy of their results however strongly depends on the quality of a wide range of input data (meteorological fields, boundary conditions, land use data and pollutant emissions) (Im et al., 2018, Zhu et al., 2019, Dufour et a., 2021, de Meij et al., 2009, de Meij et al. 2018, Cuvelier et al., 2010, Thunis et al., 2007). Many previous studies have shown that emissions are the input that has one of the most critical influences on the results of air quality modeling and, in particular, on the urban-scale source apportionment used to design air quality plans (Kryza et al., 2015, Zhang et al., 2015). More alarmingly, some studies have shown that significant uncertainties affect emission inventories, which may impeach conclusions based on air quality model results (Trombetti et al., 2018, Markakis et al., 2015). These uncertainties result from the need to compile a wide variety of information to develop an emission inventory. Indeed, these inventories are prepared for many pollutants ($NO_x$, PM, VOC, $SO_2$, CO, $CO_2$...) and for

many activity sectors (transport, industry, residential, agriculture, natural sources...) that entail different emission processes. The spatial and temporal distribution of emissions is typically based on proxies that can be estimated by very different methods. For example, top-down approaches start from emission estimates at large scale (e.g. national inventory) and disaggregate spatially and temporally the emissions with finer scale proxies. Bottom-up approaches compute directly the emissions starting from local spatial and temporal proxies based on accurate locations or high-resolution shape patterns (road, ship routes, high-resolution land use, vehicle counting, etc...). Various sources of proxies are reported to create very high resolution inventories at the urban scale (Zheng et al, 2021, Geng et al., 2017, Ramacher et al., 2021). One of the most important challenges in compiling local scale emission inventories is to remain consistent with data provided by national inventory while providing satisfactory accuracy at all locations and times. For all these reasons, compiling emission inventories can lead to different results depending on the methods and data used.

In previous works, Thunis et al. (2016) proposed a methodology to compare two emission estimates over a given area based on a limited input, the total emission per pollutant and macro sector. With this method, the differences between the total emissions of the two inventories are apportioned in terms of emission factors and activity differences. This information can then be used by emission inventory developers to identify the main causes for these discrepancies and likely errors in their estimates. However, this method is able to apportion differences between emission factors and activity only when the difference in emission factors is known for at least one of the emitted pollutants. Since this was leading to arbitrarily treat one pollutant, Clappier and Thunis (2020) improved the method by implementing a probabilistic approach to find the most likely allocation between emission factor and activity to remove this limiting assumption. In their work, Trombetti et al. (2018) then extended the approach to multiple inventories.

While the proposed approach shares some graphical representation with the previous, it differs in terms of diagnostics. The three original features of the new approach are: 1) differences in total emissions are allocated into three key components that provide information on the sectoral and spatial shares of the emissions at two geographical scales for each pollutant, 2) the capability to perform the analysis simultaneously for a large number of locations while systemically excluding emissions that are not relevant (lower emissions compared to others) and 3) to rank the largest inconsistencies between the two inventories.

This new method can be applied to two inventories that can either be two versions or two different years of a given inventory, two inventories based on distinct sources of information, e.g. CAMS-REG (Kuenen et al. 2021) and EDGAR (Crippa et al. 2018), top-down vs. bottom-up approaches, regional vs Europe wide, etc. Here, we illustrate the use of the proposed method by focusing on comparisons between two versions of the same inventory and apply the screening methodology to a continental scale inventory used to compile air quality modelling at the urban scale: the CAMS-REG inventory (Kuenen et al. 2021). In a follow-up paper, we extend and apply the approach to the comparison of different inventories.

The paper starts with a description of the screening approach that includes its required input data, the methodology itself and its output. An application with the EU-wide CAMS-REG inventory is then presented in Section 3 while further considerations are addressed in Section 4.

## 2. The screening approach

The approach presented in this article aims at comparing two emission inventories over a series of geographical areas within the domain they spatially cover. These geographical areas include two groups characterised by different scales: large (e.g. country) and focus (e.g. cities) areas. For each pollutant, the method screens the consistency of the inventories around three aspects: (1) the total pollutant emissions assigned at large scale; (2) the way these total pollutant emissions are shared in terms of sector of activity and 3) the way large scale emissions are distributed to the focus areas.

In other words, the screening approach intend to answer the following questions:

- Are there inconsistencies in total pollutant emissions at large scale level?
- Are there inconsistencies in the sectoral contributions to the emissions at large scale level?
- Are there inconsistencies in the way inventories distribute large-scale emissions spatially?

Note that the method proposed here is designed with a focus on the spatial dimension. Other uncertainties related to emission inventories (e.g. speciation of VOC or PM, temporal distribution of the emissions) are not considered.

## 2.1  Input data

Based on a 0.1 x 0.1 degree gridded emission inventory detailed in terms of emitted pollutants (denoted as "*p*") and sectors of activity (denoted as "*s*"), the data required for each pollutant and sector (denoted as a [*p,s*] couple) are

twofold and consist of:

- Emission totals aggregated over specific areas of interest (e.g., urban areas, agriculture intensive areas, industrial areas…). These areas, referred to as *focus areas*, can be freely selected and represent locations where we wish to assess the consistency of the inventory. The associated emissions are denoted by a

lowercase notation $e_{p,s}$.

- Emission totals aggregated over larger areas (e.g., country, regions, modelling domains…). In general, these areas correspond to the larger scale at which data are reported. For example, country is the scale of interest for EU wide inventories as this is the scale at which national emission totals are typically reported

or estimated. These areas are referred to as *larger scale areas* and its emissions denoted by an uppercase notation $E_{p,s}$.

The number of focus areas is denoted by N. We will also denote sectorial emission totals by an overbar ($\bar{E}_p = \sum_s E_{p,s}$).

## 2.2  Methodology

The number of [*p,s*] points under screening is equal to the product of the number of pollutants by the number of sectors itself multiplied by the number of focus areas (i.e. $N \times N_p \times N_s$). Because this number may become overwhelming, we proceed with a number of steps that help focusing the screening on priority aspects. To this end, threshold parameters are set to restrict the screening to relevant emissions (i.e., emissions that are large enough), and priority is a second step given to the detection of the largest differences between inventories among those relevant

data. These steps, schematically represented in the flowchart of Figure 1, are discussed in the next sub-sections.

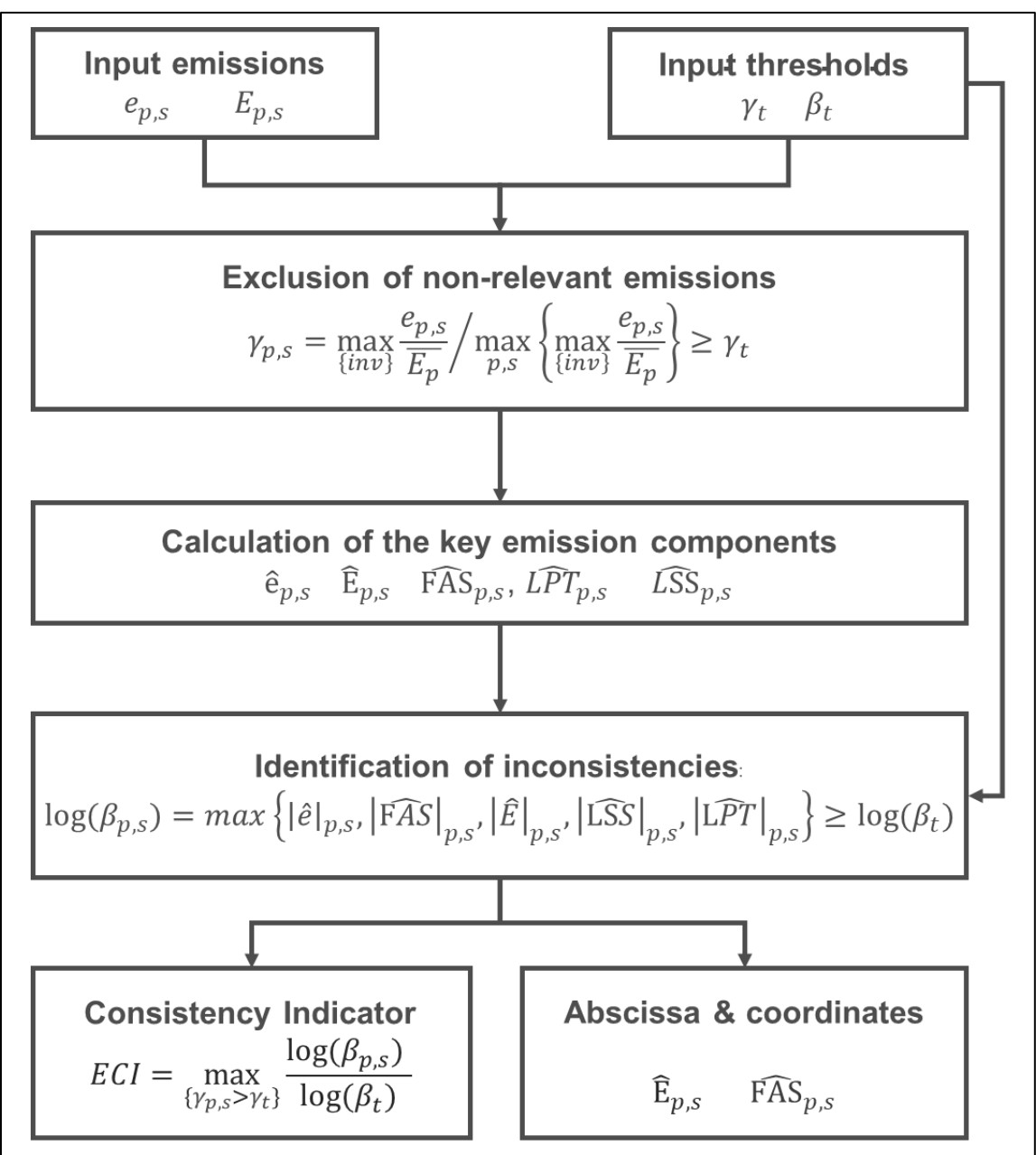

*Figure 1: Flow chart of the steps to screen emission inventories. See details in the text.*

### 2.2.1    Exclusion of non-relevant emissions

Not all emission data (*E* and *e* for all pollutant and sector) are kept for the analysis because large differences on small numbers are not relevant. For the exclusion, we proceed as follows. For each [*p,s*] and each inventory, we

calculate the ratio between the focus area emission ($e_{p,s}$) and its respective larger scale pollutant total, i.e. $\overline{E_p}$.

$$X_{p,s} = \max_{\{inv\}} \frac{e_{p,s}}{\overline{E_p}} \qquad (1)$$

For each [*p,s*] couple, the maximum values of the two inventories is then normalized by the maximum obtained over all [*p,s*] and over the two inventories for a given focus area.

$$\gamma_{p,s} = \frac{X_{p,s}}{\max\limits_{p,s}\{X_{p,s}\}} \qquad (2)$$

This final ratio informs on the relative importance of each [p,s] couple. Then, [p,s] couples are excluded from the analysis when this ratio is below a given threshold value $\gamma_t$ (arbitrary input). The purpose of this scaling is to avoid flagging issues for emissions that are proportionally less relevant. Rewriting (1) as $X_{p,s} = \max\limits_{\{inv\}} \frac{e_{p,s}}{\bar{e}_p} \frac{\bar{e}_p}{\bar{E}_p}$, we see that we exclude from the analysis:

1.  sectorial activities that have a low share over 'focus areas' (low $\frac{e_{p,s}}{\bar{e}_p}$) e.g., NMVOC emissions in the power plants sector, and/or

2.  emissions for pollutants that have a low urban share (low $\frac{\bar{e}_p}{\bar{E}_p}$), e.g., agriculture.

As we will see in the application (Section 3), this exclusion step leads to eliminating a large majority of the [p,s] couples from the screening process (between 80 and 90%).

### 2.2.2    Decomposition into key components

The objective of the decomposition is to isolate emission characteristics that are associated to the inventory compilation process, in order to facilitate the resolution of the detected inconsistencies. The three main characteristics are (1) the Pollutant Totals over Large areas (*LPT*), the Sectorial Shares over Large areas (*LSS*) and the Activity Share over the Focus areas (*FAS*). To isolate these components, we decompose the ratio of the known pollutant-sector emissions for each focus areas as follows:

$$\frac{e_{p,s}^1}{e_{p,s}^2} = \frac{\frac{e_{p,s}^1}{E_{p,s}^1}}{\frac{e_{p,s}^2}{E_{p,s}^2}} * \frac{\frac{E_{p,s}^1}{\bar{E}_p^1}}{\frac{E_{p,s}^2}{\bar{E}_p^2}} * \frac{\bar{E}_p^1}{\bar{E}_p^2} \qquad (3)$$

where $\bar{E}_p$ represents the larger scale emissions summed over all sector for a given pollutant. Superscripts refer to the two inventories used for the screening. Equation (3) is an identity where all terms are known from input quantities, i.e. the focus and larger scale emissions detailed in terms of pollutants and sectors. The three terms on the right-hand side of the identity provide information on the *FAS*, *LSS* and *LPT*, respectively.

For convenience, we rewrite equation (3) in logarithm form as:

$$log\left(\frac{e_{p,s}^1}{e_{p,s}^2}\right) = log\left(\frac{\frac{e_{p,s}^1}{E_{p,s}^1}}{\frac{e_{p,s}^2}{E_{p,s}^2}}\right) + log\left(\frac{\frac{E_{p,s}^1}{\bar{E}_p^1}}{\frac{E_{p,s}^2}{\bar{E}_p^2}}\right) + log\left(\frac{\bar{E}_p^1}{\bar{E}_p^2}\right) \qquad (4)$$

Which can be rewritten as equation (5) with simplified notations:

$$\hat{e} = \widehat{FAS} + \widehat{LSS} + \widehat{LPT} \qquad (5)$$

where the hat symbol indicates that quantities are expressed as logarithmic ratios. These three quantities are at the basis of the screening methodology and serve as input for the graphical representation as well.

### 2.2.3    Identification of inconsistencies

One of the main steps of the methodology consists in keeping only the largest differences among the relevant emissions identified in Section 2.2.1. The comparison of two emission inventories always leads to different estimates, as inventories can be calculated by different methods (e.g. different activity data, emission factors and spatial disaggregation to the grid).

Differences originate from methodological choices but also from errors generated during the inventory compilation process. When differences are small, it is not possible to tell whether they originate from methodological choices or from errors. Moreover, it is not possible to assess whether one methodological choice leads to an improvement as compared to the other, because true emissions remain unknown (Kryza et al., 2015). We will refer to these small differences as "uncertainty".

Although very large differences may result from methodological choices as well (e.g. inclusion or not of condensable emissions for the residential sector), they are more likely to be associated to errors. Given the magnitude of the differences, it will in most cases be possible to identify one best value out of the two inventory estimates, even though the truth is unknown. These large differences therefore point to a list of potential issues for inventory compilers to check and fix where applicable, opening the way to potential improvement. In this work,
these large differences are named "inconsistencies" and are intended as differences that are large enough to ensure that one of the two inventory values is tenable (i.e. justifiable) whereas the other is not.

In the proposed screening methodology, a threshold is introduced to distinguish inconsistencies from uncertainties. This arbitrary level is denoted as $\beta_t$ and is a free input data in this screening process.

The detection of inconsistencies is performed as follows. For each [p,s] we check that any of the two input data: $\left|\hat{E}\right|$ and $|\hat{e}|$ do not exceed the threshold but also that any of the three main components: $\left|\widehat{FAS}\right|$, $\left|\widehat{LSS}\right|$ and $\left|\widehat{LPT}\right|$ do not either. This is to flag potential compensating effects between $\widehat{FAS}$ and $\hat{E}$ since $\hat{e} = \widehat{FAS} + \hat{E}$, and between $\widehat{LPT}$ and $\widehat{LSS}$ since $\hat{E} = \widehat{LPT} + \widehat{LSS}$. To achieve this, the following indicator is defined.

$$log(\beta_{p,s}) = max\left\{|\hat{e}|_{p,s}, \left|\widehat{FAS}\right|_{p,s}, \left|\hat{E}\right|_{p,s}, \left|\widehat{LSS}\right|_{p,s}, \left|\widehat{LPT}\right|_{p,s}\right\} \tag{6}$$

Differences beyond the threshold ($\beta_{p,s} \geq \beta_t$) are then flagged as inconsistencies.

### 2.2.4 Calculation of an emission consistency indicator (ECI)

As a follow-up step, all [p,s] couples that remain after the relevance ($\gamma_{p,s} > \gamma_t$) and inconsistency detection steps ($\beta_{p,s} > \beta_t$), are used to calculate an "Emission Consistency Indicator (ECI)" as follows:

$$ECI = \max_{\{\gamma_{p,s}>\gamma_t\}} \frac{\log(\beta_{p,s})}{\log(\beta_t)} \tag{7}$$

The *ECI* quantifies the maximum difference among all relevant [p,s], normalized by the inconsistency level ($\beta_t$). It therefore quantifies the ratio between the maximum inconsistency and the assumed level of uncertainty. A value of *ECI* less than one means that all differences are considered as uncertainty (in other words none of the inventory can be identified as best performing). Together with the *ECI*, which quantifies this maximum difference, we associate
the percentage of inconsistent [p,s] with respect to the total number of relevant data, to provide information on the number of detected inconsistencies. To facilitate the screening process, these concepts are displayed graphically. This is discussed in the next Section.

### 2.3 Graphical display

#### 2.3.1 Diamond diagram
For the graphical representation, we use an aggregated form of equation (5), recalling that the two last terms on the right hand side combine into the ratio of the larger scale [p,s] emissions, i.e.:
$$\hat{e} = \widehat{FAS} + \hat{E} \tag{8}$$
where FAS is related to the large scale-to-focus scale emission share and E is related to the large scale emissions. Relation (8) is the basis of the "diamond" diagram (Figure 2) that provides an overview of all inconsistencies detected during the screening process. In this diagram, each inconsistent emission [p,s] is represented by a point that
has larger scale emissions ($\hat{E}$) as abscissa and focus activity share as ordinate ($\widehat{FAS}$). The sum of these two terms ($\hat{e}$) is equal for points that lie on "−1" slope diagonals. At this stage it is important to note that positive differences in terms of larger scale emissions and focus area shares will characterize points lying on the right and top parts of the

diagram, respectively. In addition, the upper right and lower left diagram areas indicate summing-up effects whereas the lower right and top left areas highlight compensating effects.

The diamond shape (in the middle of the diagram) derives from equation (8) where the $\beta_t$ threshold is used to draw the inconsistency limit for each of its three terms. Each [*p,s*] point lying outside this shape is therefore characterized by an inconsistency in terms of either *E*, *FAS* or/and *e*, small or large according to its relative position in the diagram. The calculation of the inconsistency limit (equation 6) however considers *LPT* and *LSS* as two additional

235 criteria. Because of their link ($\widehat{E} = \widehat{LPT} + \widehat{LSS}$), a point within the diamond represents therefore an inconsistency in terms of *LPT* and *LSS* that compensate each other, since their sum remains lower than the threshold ($\widehat{E} \leq \log(\beta_t)$, otherwise, the point would lie outside the diamond). We recall that LPT is related to the total of the large scale emissions whereas LSS provides information on their sectoral share.

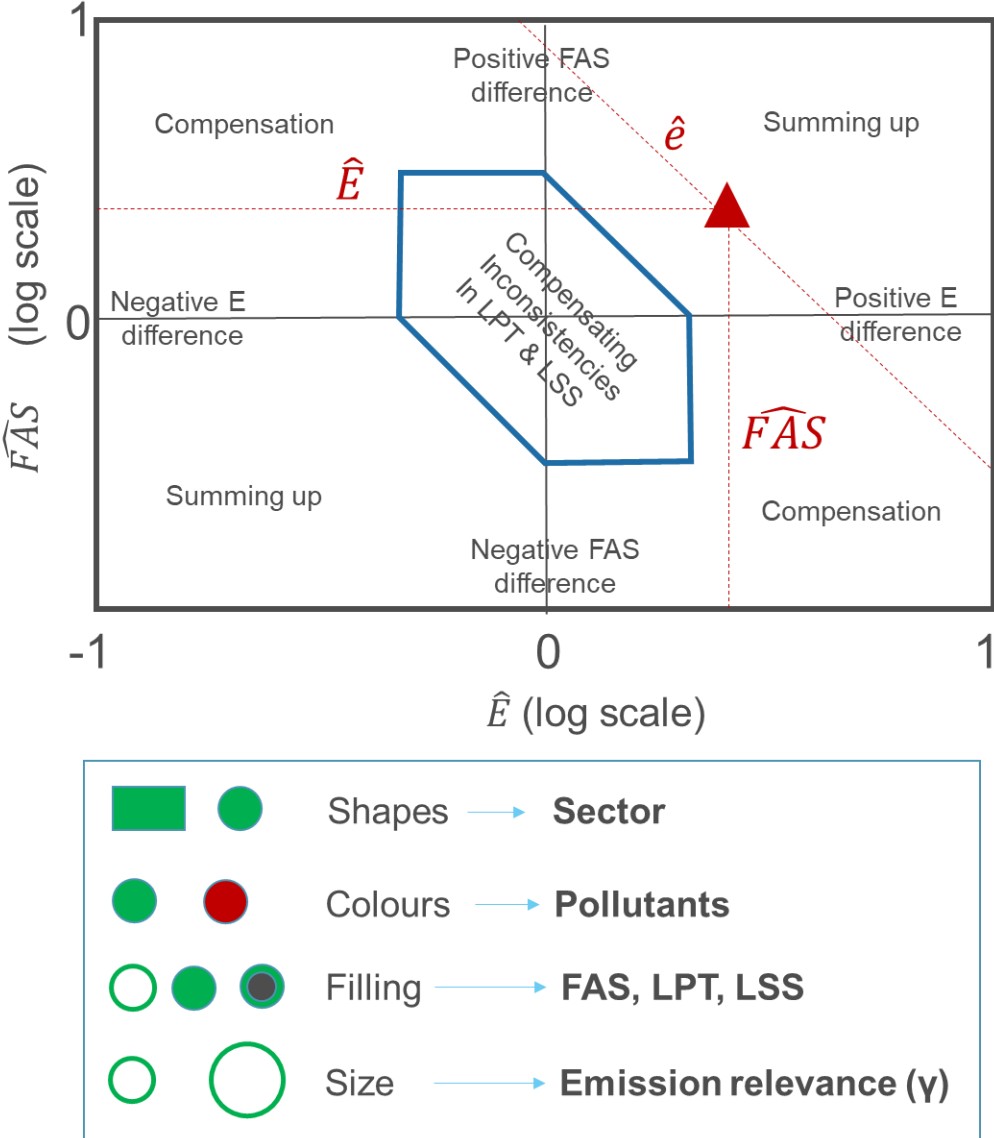

*Figure 2: Diamond diagram and representation of the codification. LPT provides information on the total of the large scale emissions, LSS on their sectoral share, FAS on the large-to-focus scale share and E on the large scale emissions*

In this diamond diagram, shapes are used to differentiate activity sectors while colours differentiate pollutants. The size of the symbol is proportional to the relevance of the emission contribution ($\gamma$). Finally, we use symbol filling to distinguish priorities among inconsistencies related to the three components: *LPT*, *LSS* and *FAS*. The priority is set as follows: $1 - LPT$, $2 - LSS$ and $3 - FAS$. This is motivated by the fact that larger scale inconsistencies are easier to tackle and might correct for many focus area inconsistencies at the same time (i.e. for all focus area belonging to a given larger area). The priority is then set by checking in this order if the component exceed the threshold or is larger than the remaining components. In practice, this is implemented as follows:

1. If $\widehat{LPT} \geq \log(\beta_t)$ or $\widehat{LPT} \geq max[\widehat{FAS}, \widehat{LSS}]$, then priority is set to the larger scale pollutant total (*LPT*).
2. If step 1 is not fulfilled and if $\widehat{CSS} \geq \log(\beta_t)$ or $\widehat{CSS} \geq \overline{UAS}$ then priority is set to the larger scale sectorial share (*CSS*)
3. If neither steps 1 and 2 are fulfilled, priority is set to the focus area activity share (*FAS*)

Note that compared to the emission diagram proposed by Thunis et al. (2016) and Clappier and Thunis (2020), the diagram proposed here does not distinguishes between acceptable (within the diamond) and non-acceptable data (outside the diamond) but displays only inconsistencies (i.e. data to be checked for which some explanation must be found). Moreover, the current formulation does not rely on probabilistic assumptions and directly relates to emission characteristics that are readily available to emission developers.

### 2.3.2    Supporting diagrams

In addition to the diamond diagram, other diagrams are proposed to support the interpretation of the screening. These diagrams are designed to provide additional information by detailing further some aspects (e.g. geographical) at the expenses of aggregation or simplification (e.g. limitation to top inconsistencies) on other aspects. These diagrams are:

- *Overview map*: Data are displayed on a geographical map to easily identify the inconsistencies for each focus area. However, only the maximum inconsistency ($\max_{p,s}\{\beta\}$) for each focus area is shown. While the size is here proportional to the magnitude of the inconsistency, the symbol shapes, colours and filling remain similar to the overview diamond.

- *Barplot*: For a given pollutant and focus area, this diagram allows visualizing inventory differences directly in terms of *FAS*, *LSS* and *LPT* components. This diagram is used here as validation means (with respect to the diamond results).

We discuss further these visualizations in the application section, showing a graphical example and comments for each type of plot.

### 3.    Application

### 3.1  Input
In this section, we apply our screening methodology to the CAMS-REG regional anthropogenic emission inventory that covers emissions for UNECE-Europe for the main air pollutants and greenhouse gases (Kuenen et al., 2021). The method starts from the reported emissions by European countries to UNFCCC (for greenhouse gases) and to EMEP/CEIP (for air pollutants) and have been aggregated into 246 different combinations of sectors and fuels. Reported data are analysed by sector and completed with alternative emission estimates where the completeness, consistency and/or quality of the reported data was not sufficiently accurate. In practice, reported data were found fit for purpose for EU Member States and the UK, Norway, Iceland and Switzerland, while for other countries alternative emission estimates were used. In addition, some further modifications were made to the dataset for which we refer to Kuenen et al. (2021). This results in a complete emission inventory for all countries, which is then spatially distributed at high resolution using a consistent methodology over the whole domain.
For the comparison presented in this paper, we use two versions of the CAMS-REG inventory for the same year 2015:

- CAMS-REG v2.2.1: a version covering the years 2000-2015, based on the official reported data of air pollutants and greenhouse gases in the year 2017;
- CAMS-REG v4.2: this version covers the years 2000-2017, based on the official reported data of air pollutants and greenhouse gases in the year 2019.

In comparison to version 2.2.1, the methodology for the total emissions by country, pollutant, sector and year in CAMS-REG-v4.2 is largely similar, however the difference in reporting year of emissions may introduce substantial differences in sectoral or country total emissions in specific cases. This happens since official country emission data are reported on an annual basis for all historical years, which implies that emissions for the year 2015 have been reported annually starting from 2017. Inventories are continuously improved by means of updating methodologies (new activity data, new emission factors, etc.) which implies that historical emissions from a single year (2015 in this case) are different in each reporting year. On the other hand, some of the changes are the result of methodological changes in the CAMS-REG inventory, which are mainly related to the spatial distribution, where CAMS-REG-v4.2 contains among others an improved split in road transport emissions between urban, rural and highway shares, agriculture (new proxies for manure spreading, fertilizer application based on JRC CAPRI model (Britz et al. 2015)). It also uses a new approach for Agricultural Waste Burning, improves the point source database, and uses updated harmonized inland and sea shipping based on the FMI STEAM model (Jalkanen et al., 2009). Further details on these changes are provided in Kuenen et al. (2021).

It is important to stress that the proposed screening methodology assesses the overall consistency of the two inventory versions, i.e. it covers the consistency of the inventory compilation itself but also the consistency of its input data (in this case: country reported emissions).

Focus areas are defined as the functional urban areas (FUA, OECD 2012) for which emissions ($e_{p,s}$) are obtained by aggregating grid cell values. The FUA is composed of a core city plus their wider commuting zone, consisting of the surrounding travel-to-work areas. 150 FUAs across Europe, depicted with blue contours in Figure 3 are selected for this screening. Details on these cities are provided in Thunis et al. (2018). The larger scale emissions ($E_{p,s}$) are defined at country level, level at which CAMS-REG takes the official reported data as input.

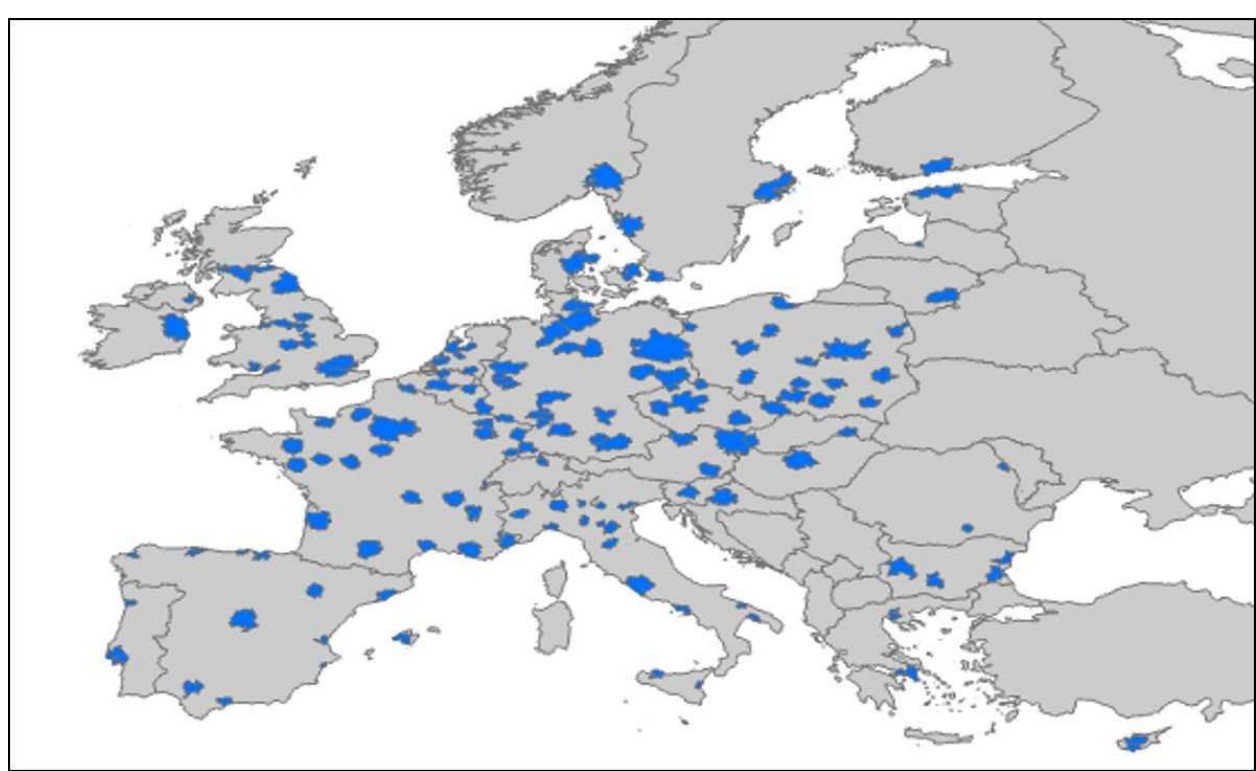

*Figure 3: Map of the Functional Urban Areas (FUA) considered for the emission screening.*

In terms of pollutants, $E_{p,s}$ and $e_{p,s}$ include the following: $NO_x$, NMVOC, $PM_{2.5}$, $PM_{co}$ (coarse PM, calculated as the difference between $PM_{10}$ and $PM_{2.5}$ emissions from CAMS-REG), $SO_2$ and $NH_3$, whereas sectors are based on the Gridded Nomenclature For Reporting (GNFR) classification. The original GNFR sectors have been aggregated in 5 categories: road transport (F), residential (C), power plants (A), industry (B) and others (D-E-I-J). The latter category includes waste, fugitive emissions, solvents and off-road transport. As we focus on urban areas, agriculture emissions (K-L) are not relevant and therefore not included as part of the analysis. Shipping (G) and air transport (H) are also not part of the analysis.

For this inventory, *LPT* concerns the country level total emissions, *LSS* the sectoral total emissions at country level and *FAS* the sectoral emissions at the local level (city of interest). Note that the selection of the larger scale and focus areas, as well as of the sectors to consider is a free user choice.

Finally, the relevance and inconsistency threshold are set in this work to $\gamma_t = 0.5$ and $\beta_t = 2$. Although the choice of these threshold is arbitrary and may seem challenging, this is not the case in practice and identifying the inconsistencies is a robust process. A too low threshold will lead to detecting too many differences among which the smallest (at uncertainty level) do not allow assessing what is best. However, the largest differences (inconsistencies) are yet identified and can be taken care of to improve one or both inventories. A too low threshold will therefore lead to confusing information by mixing uncertainties and inconsistencies. On the other hand, a too high threshold will lead to detecting too few inconsistencies and therefore to missing errors that could potentially be corrected. In practice, it is recommended to start with a high threshold and lower it progressively until differences cannot be justified anymore. The values for the relevance and inconsistency threshold ($\gamma_t$ and $\beta_t$) presented here reflect these considerations.

### 3.2 Results

The diamond diagram (Figure 4) displays the inconsistencies between the CAMS-REG-v2.2.1 and v4.2 inventories over the entire emission domain, i.e., Europe. Among the 4500 screened [*p,s*] points, corresponding to the product of 150 cities per 6 pollutants and 5 sectors, only about 450 are kept after application of the relevance threshold ($\gamma \geq 0.5$). Among the remaining 450 points, 46 show inconsistencies ($\beta > 2$) as indicated by the lower right number (NI). In other terms, about 9% of the relevant [*p,s*] show inconsistent values (number associated in brackets) aside the *ECI*. In our application, the *ECI* is about a factor 70, meaning that the maximum inconsistency is about 70 times larger than the assumed level of uncertainty.

The summary table at the bottom of the diagram supports the interpretation by ranking inconsistencies in terms of pollutant, sector and type (*FAS* vs. *LSS* vs. *LPT*). Most of the inconsistencies arise for $SO_2$ (13) and $PM_{co}$ (16), primarily from the industrial and residential sectors, and originate both from the country (LPT+LSS=2+22) and local scales (FAS=22). It is worth noting that few inconsistencies are within the diamond, pointing to a limited number of compensations between *LPT* and *LSS* large scale inconsistencies (i.e., overestimation in terms of pollutant total compensated by an underestimation in terms of sectorial share, and vice versa). This information is graphically detailed by representing the [*p,s*] points with varying colours and symbols. $PM_{co}$ inconsistencies are important in magnitude and are mainly related to lower estimates (points are below the X axis) by v4.2 of the urban share (points are distributed along a vertical line). It is interesting to note that points are mostly distributed either on a vertical or horizontal line indicating the absence of compensation (top right and bottom left part of the diagram) or summing-up effects (top right and bottom left part of the diagram).

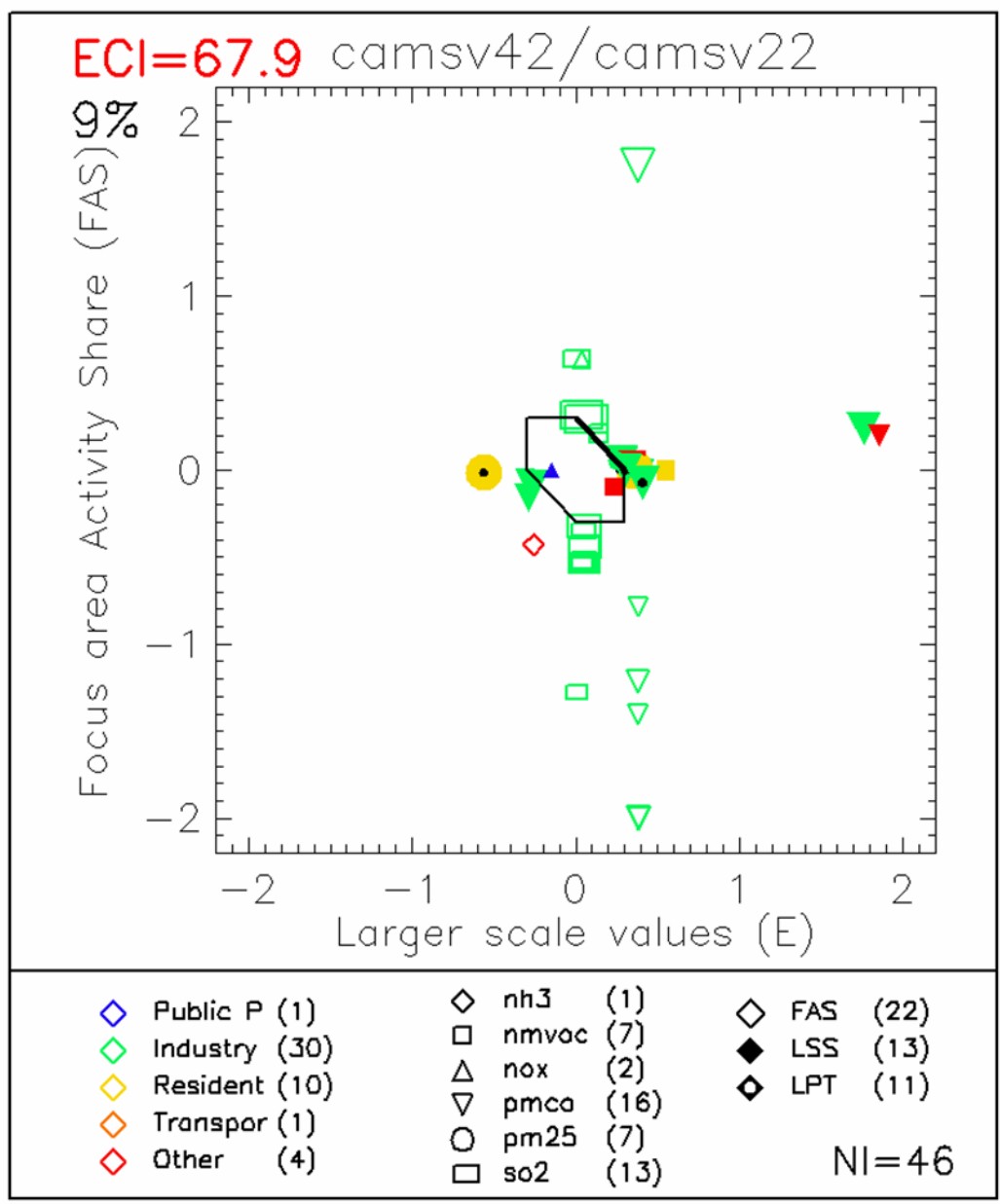

Figure 4: Overview diamond. Symbols and colours are used to distinguish pollutants and sectors, respectively. The X- and Y-axis indicate inconsistencies in country emissions ($\hat{E}$) and urban share ($\widehat{FAS}$), respectively while the overview table provides information on the origin of the inconsistencies in terms of sector, pollutant and country/city scale (FAS, LSS, LPT). The ECI indicator indicates the ratio between the magnitude of the maximum inconsistency and the assumed level of uncertainty (β). The percentage number indicates the fraction of inconsistencies ($\beta \geq \beta_t$) among the relevant emissions ($\gamma \geq \gamma_t$). See additional explanations in the text.

The European map presented in Figure 5 flags out the inconsistency that dominates in each city. It is interesting to note that while the total number of inconsistency (46) might seem large, the map shows that in some countries, the same type of inconsistency is widespread. This is the case of the $PM_{co}$ industrial emissions in the UK, the $SO_2$ industrial emissions in France, or for the NMVOC residential emissions in the Czech Republic. As the size of the symbol is here proportional to the magnitude of the inconsistency, the $PM_{co}$ emissions from the industrial sector might need a priority check in some UK cities. Even though these inconsistencies appear in different countries, their type is similar and resolving one might bring useful information to resolve the others.

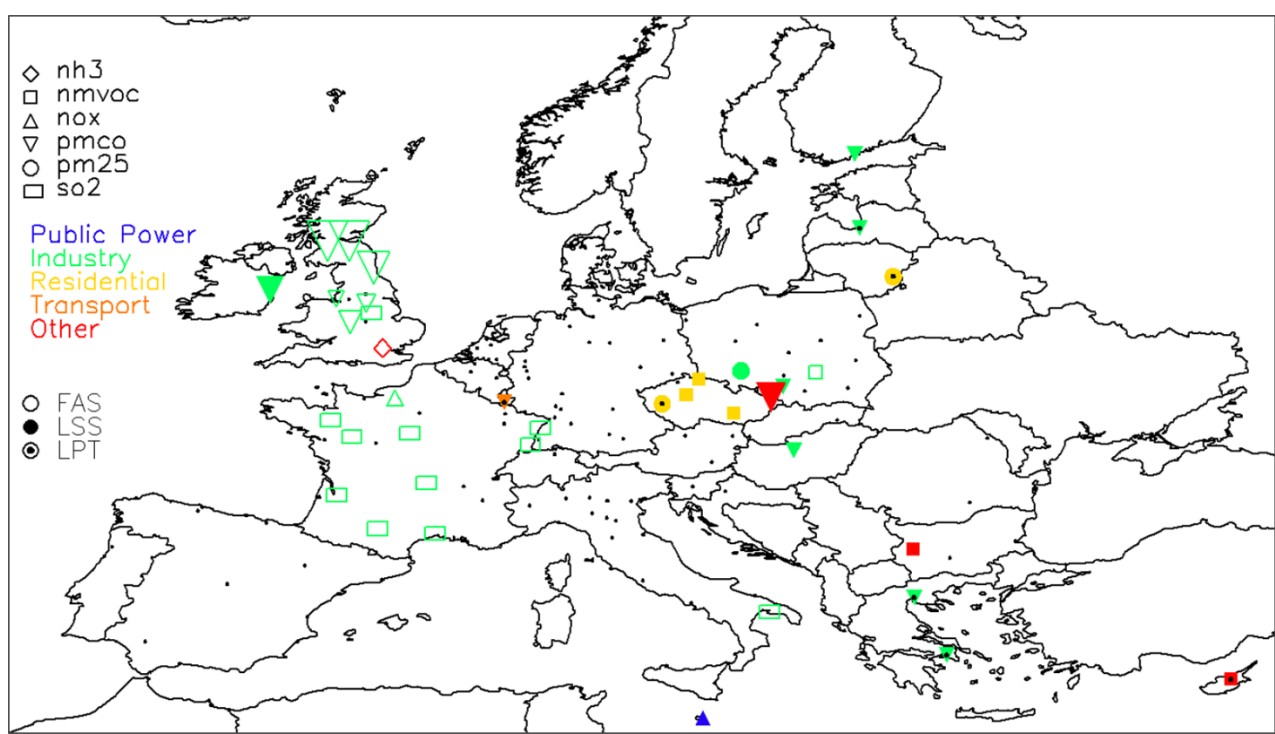

*Figure 5: EU (overview) map. Only cities where at least one [p,s] couple ratio with relevant emissions ($\gamma \geq \gamma_t$) is above the inconsistency threshold ($\beta \geq \beta_t$) are shown by a symbol. If more than one [p,s] fulfils these two conditions, only the largest is shown. For all others, cities are represented by a black dot.*

We focus now on some of these inconsistencies and try to understand their origin. Examples are picked among those showing important inconsistencies with the aim to illustrate different type of inconsistencies, i.e. in terms of LSS, FAS and LPT (Figure 6). Note that depending on pollutant and sector, the ECI can differ quite strongly in magnitude explaining the different range of values in our examples.

*Vilnius: inconsistent country totals for PM$_{2.5}$*: In Vilnius (Lithuania), the *ECI* is about 2 (see *diamond* plot in Figure 6 top row), indicating inconsistencies that are about twice larger than the level of uncertainty. The flagged inconsistency is aligned along the x-axis indicating issues in terms of country PM$_{2.5}$ values, in particular pollutant levels (*LPT*). The associated bar-plot highlights a factor 2 to 3 difference between the CAMS-REG-v2.2.1 and -v4.2 estimates (Figure 6). Note that the country sectorial shares are also diverging for the industrial but also transport sectors. This is however seen by the screening tool as a second priority.

The changes can be explained by the changes in the emission reporting that is used as input to the CAMS-REG inventories. Significant updates were made in the 2019 submission compared to the 2017 submission for Lithuania. For example, PM$_{2.5}$ emissions from residential sector decreased by nearly a factor 4, whereas road transport emissions increased by ~70%. National total PM2.5 emissions reported by Lithuania for 2015 were reduced by more than 50% between submissions in 2017 and 2019.

*Dublin: inconsistent industry country share for PM$_{2.5}$*: In Dublin, the *ECI* is about 50, indicating inconsistencies that are about 50 times larger than the level of uncertainty. The flagged inconsistency (PM$_{co}$ from industry) lies on the right indicating a much larger value attributed to this pollutant/sector in the CAMS-REG-v4.2 version. This is confirmed in the associated bar-plot (Figure 6 second row) that highlights a totally different industrial share in the two inventories. This country scale issue is partly echoed in the urban share, but this is seen by the screening tool as a second priority. Similar to the Vilnius case above, this can be explained by changes in country reporting. Whereas total emissions in the 2017 submission for Industry were 1.5 kt PM2.5 and 1.6 kt PM10, in the 2019 submission the PM2.5 emissions amounted to 1.9 kt and PM10 emissions were 7.7 kt. Hence PMco emissions from industrial sources were increased by more than a factor 50, from 0.1 to 5.8 kt, between both versions.

*Newcastle: inconsistent industry urban share for $PM_{CO}$*: In Newcastle (UK), the ECI is 68 (Figure 6 third row), indicating inconsistencies that are about 70 times larger than the level of uncertainty. The flagged inconsistency ($PM_{co}$ from industry) is mostly driven by the urban share but country values differ largely as well (factor 2). Note that large differences of the same type also occur for $PM_{2.5}$. While this is not flagged as a major inconsistency by the screening approach (because the relative importance of the emissions ($\gamma$) is too small), this might become the case

when the $PM_{co}$ inconsistency has been resolved. The associated bar-plot highlights differences in country totals and country sectorial shares but these are not sufficient (in terms of $\gamma$ or $\beta$) to trigger the flagging. On the contrary, the very large difference in the urban share, with CAMS-REG-v4.2 exceeding -v2.2 emissions by almost a factor 100 are flagged. As mentioned above, this inconsistency is present in many UK cities. The inconsistency can be explained partly by changes in reporting between 2017 and 2019 submissions, as the $PM_{co}$ from industry increased

from 15.8 to 35.2 kt. For the distribution in the country, E-PRTR is used for distributing emissions to point source installations. When checking in detail for this location, a factor 1000 error in E-PRTR reporting was found, which lead to an over-allocation of PM emissions from the industrial sector to this specific industrial site located within the Newcastle urban area. This makes that in CAMS-REG-v4.2 emissions in this particular location are overestimated, which is compensated for by underestimated emissions elsewhere in the UK.

*London: inconsistent "other" urban share for $NH_3$*:
In London (UK), the *ECI* is 2.5, indicating inconsistencies that are about 2.5 times larger than the level of uncertainty (Figure 6 bottom row). The flagged inconsistency ($NH_3$ from the "other" sector) results from both urban and country differences that add up but the dominating factor is the urban share. The associated bar-plot confirms

this issue while differences in country totals and country sectorial shares appear moderate in comparison to the urban share issue. In contrast to Newcastle, this issue is only appearing for London.
In this specific case, it was found that a relatively large part of $NH_3$ emissions was reported in the category "other waste" for the United Kingdom as a whole. Given the relatively low importance of the sector "other waste" and the absence of point source information for $NH_3$ for this particular sector, these emissions were allocated using a

surrogate point source distribution where all emissions ended up in the same point source in London, thus significantly over-allocating emissions in this location. This therefore points to an inconsistency in the CAMS-REG methodology.

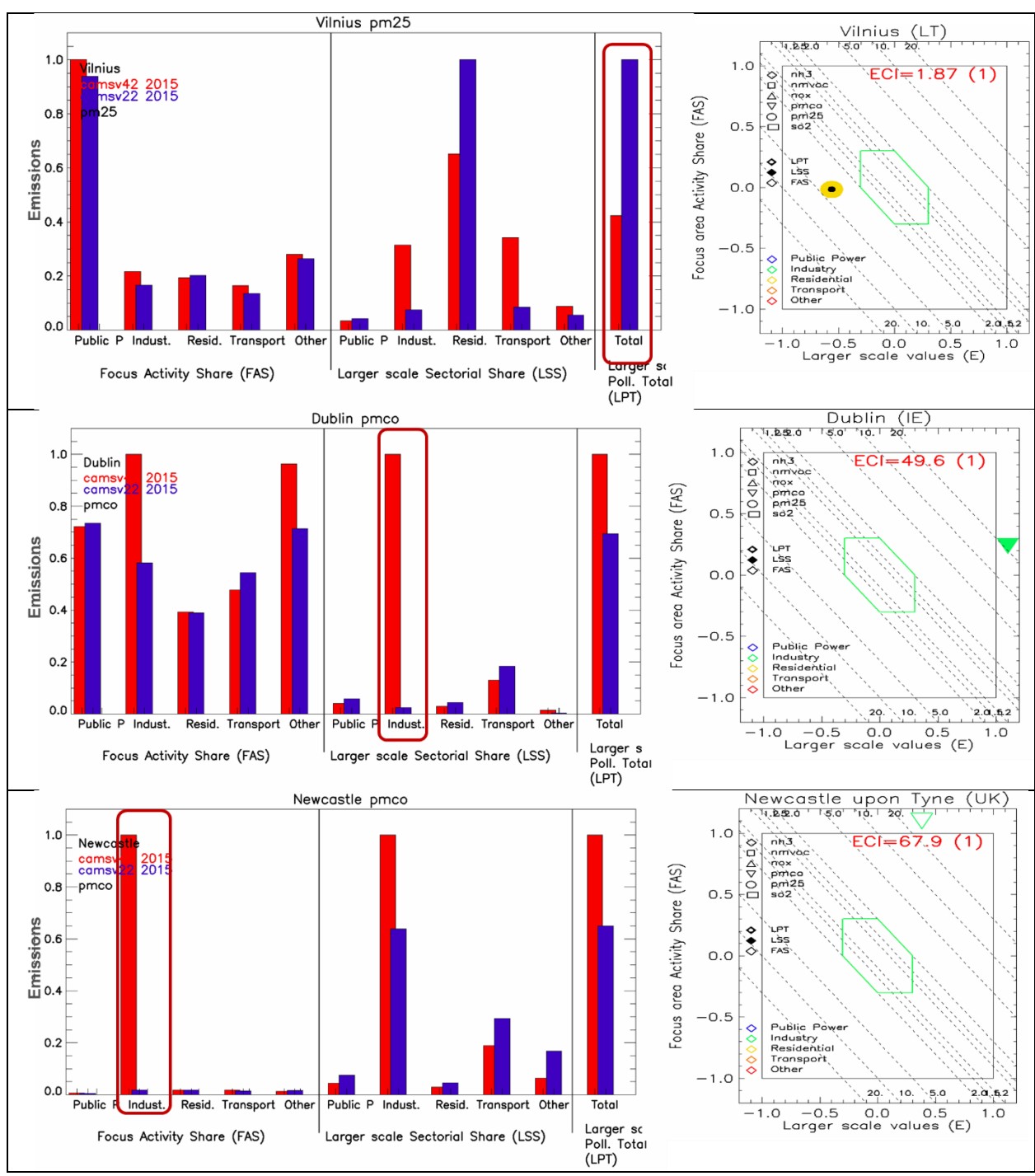

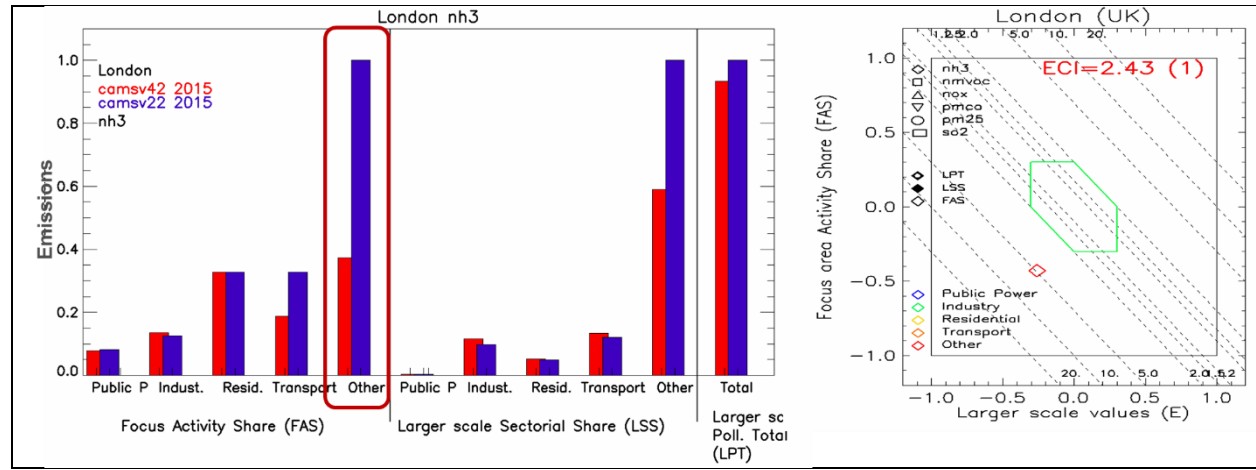

*Figure 6: Barplots and city diamond for the cities inconsistencies. The bar-plots show the values reached for the FAS, LSS and LPT by the two inventories, based on input emission values. Data are graphically scaled relatively to the maximum reached for each of the three factors (FAS, LSS and LPT).*

Half of the inconsistencies between the two versions of CAMS-REG considered in this study can be attributed to changes in country reporting. All European countries annually revise and report their historical annual emissions
back to 1990, hence the emission of e.g., 2015 are reported every year again. The differences between both versions may be the result of a correction of an error and/or the implementation of a different methodology to estimate the emissions. This may be checked in the reports (IIRs) that are submitted annually along with the reported emissions, but likely not all changes are documented in detail. This makes that only in a selection of cases it will be possible to define an error in one of the inventories as such, and hence define the better inventory.
An important driver for inventory improvement in recent years has been the annual review of air pollutant inventory data under the NEC Directive organised by the European Commission[1], which has led to substantial revisions or nationally reported data since 2017 for all pollutants and sectors.

## 4. Further considerations

The approach presented in this work is intended as a screening to flag inconsistencies. Only differences that are
above a user-defined threshold ($\beta_t$) are detected and smaller differences are disregarded. This threshold reflects the limit between relatively small differences for which no emission inventory can be estimated to be the best (because true emissions are unknown) and differences that are so large that they are likely associated with a large error in one of the two (or both) inventories (hence called inconsistencies) for which it should be possible to identify a best performing inventory.

While solving a few inconsistencies will generally lower the overall number of inconsistencies, this is however not always the case. Indeed a very large inconsistency can potentially lead to a $\gamma$ factor that is so large that all other [$p,s$] for that city would be disregarded in proportion. Once the inconsistency is solved, the new $\gamma$ estimates might lead to one or more new inconsistencies to be flagged. This is therefore a step-wise approach.

The settings used in this work, i.e., the choice of 150 urban areas and the country level as larger scale have been arbitrarily fixed. The methods allow for flexible choices and could be applied to other areas than urban (e.g., high emission industrial or intensive agriculture areas) to assess the consistency with respect to other types of emissions. Similarly, the larger scale can be adapted to the specific inventory and focus on regions rather than countries or have
it defined as the entire modelling domain.

The proposed application focuses on the comparison of two versions of a specific inventory (here CAMS-REG). Although more challenging, the screening method can be applied to the comparison of two different inventories. Obviously, additional challenges will appear, in particular (1) differences in terms of spatial resolution that might

---

[1] https://ec.europa.eu/environment/air/reduction/implementation.htm

result in sources being excluded from some grid cell for one inventory and included in the other, resulting in artificial differences or (2) the need of harmonization of the emissions in terms of sectorial categories as a first step before the comparison. This inter-comparison of inventories is the subject of a follow-up paper, where these specific issues are discussed.

Given its flexible settings, the screening method also applies to bottom-up inventories. These can then be compared with themselves (e.g. 2 versions) or with other inventories. As mentioned earlier, the smaller areas of interests can be designed at own convenience and this is also the case for the larger scale that in the extreme case can be set to the total domain area.

Finally, the screening tool also provides text information that summarizes the inconsistencies by detailing the city, sector, pollutant, type and amplitude for each of them. A comment line is associated to each of them in order to keep track of steps taken to resolve them (or not).

### 5. Conclusions

In this work, we proposed and discussed a screening method to compare two emission inventories. The overall goal
is to improve the quality of emission inventories by feeding back the results of the screening to inventory compilers who can check the inconsistencies found and where applicable resolve errors. The method targets three different aspects: 1) the total emissions assigned to a series of large geographical area, countries in our application; 2) the way these country total emissions are shared in terms of sector of activity and 3) the way inventories spatially distribute emissions from countries to smaller areas, cities in our application. The method provides a way to quantify the level
of consistency (intended here as a whole, i.e. emission compilation plus all input relevant data) between two inventories.

Given the large and possibly overwhelming amount of data to analyze (many pollutants, activity macro sector and cities), the first step of the screening approach consists in sorting the data for the comparison and keep only emission
contributions that are relevant enough. In a second step the method identifies, among those relevant emissions, the most important differences that are evidence of methodological divergence and/or errors that can be found and resolved in at least one of the inventories. Although this screening does not allow to check the quality of the inventories in an absolute way, the magnitude of the differences is often large enough that it is possible to identify one best value out of the two inventory estimates, even though the truth is unknown.

The approach has been used to compare two versions of the CAMS-REG European scale inventory over 150 cities in Europe for selected activity sectors. The versions 2.2.1 and 4.2 of this inventory differ both in terms of reporting year (new activity data, new emission factors, etc.) and in terms of spatial distribution (e.g. split in road transport emissions between urban, rural and highway shares, new proxies for agriculture…). Among the 4500 screened
pollutant-sectors, about 450 were kept as relevant among which 46 showed inconsistencies. The analysis indicated that these inconsistencies were almost equally arising from reporting by countries and methodological issues in CAMS-REG (e.g. spatial distribution). They mostly affect $SO_2$ and PM coarse emissions, from the industrial and residential sectors. Differences in terms of reporting may be the result of a correction of an error and/or the implementation of a different methodology to estimate the emissions. But the fact that about half of the
inconsistencies can be attributed to changes in country reporting stresses the necessity to further checking the informative inventory reports (IIRs) that are submitted annually along with the reported emissions. For inconsistencies related to the CAMS-REG methodology and in particular the spatial distribution therein, the analysis presented here showed that for specific cities, screened errors could be explained, and some of them resolved, leading to improved inventories.

Although only a particular example has been discussed here, the screening approach is general and can be used for other types of applications related to emission inventories. The approach can be applied to other inventory scales (e.g. regional or local) and can be tuned to address different sectors or areas. Intensive agriculture or industrial areas could for example be added to the urban agglomerations considered in this work. Because the type of emission
expertise strongly depends on the type of applications, the expertise relevant to a given application is necessary to analyze the screening results and correct likely errors. The screening approach also allows assessing the consistency of a temporal series of emissions, or compare inventories based on different source of information or even compare inventories based on different methodologies (e.g. comparison of top-down and bottom up) are possible. The latter are the subject of a follow-up paper.

## 6. Appendix

This section provides details on the cities considered in this study, in terms of sectors and emitted pollutants for which a distinction is made between non-relevant, relevant and inconsistent emission inventory pairs. The table below distinguishes for each city between non-relevant and relevant, the latter being further split into consistent and inconsistent.

*Table 1: Details of the screening output for the 150 cities, 6 emitted pollutants (NH$_3$, NMVOC, NO$_x$, PM$_{co}$, PM$_{2.5}$ and SO$_2$) and 5 considered sectors (Public Power, Residential, Transport, Industry and Other). In the table, sectors are represented by their first letter. Empty cells mean that no relevant emissions has been screened. Black letters indicate relevant and consistent emissions whereas red letters indicate relevant but inconsistent emissions. Grey shaded cells indicate that at least one sector is inconsistent for the considered city-pollutant pair.*

| | NH3 | NMVOC | NOX | PMCO | PM25 | SO2 |
|---|---|---|---|---|---|---|
| Cardiff | | O | | | R | |
| De Hage | | O | | | | |
| Coruna | | | | | | I |
| Alicante | | O | T | I | R | |
| Amsterdam | | O | T | I | I | PI |
| Angers | | O | T | I | | I |
| Antwerpen | | | | | | I |
| Arhus | | O | | R | I | |
| Athina | | TO | | I | | I |
| Augsburg | | O | I | | | |
| Barcelona | | O | | | | |
| Bari | | O | T | | R | I |
| Belfast | | O | | | | P |
| Berlin | | O | I | | | |
| Bialystok | | O | T | R | R | P |
| Bilbao | | O | | | | I |
| Bologna | | O | T | I | R | I |
| Bonn | | O | I | | | |
| Bordeaux | | O | T | I | | I |
| Bratislava | | O | IT | | R | |
| Braunschweig | | | | | | PI |
| Bremen | | O | T | I | | PI |
| Brescia | | O | T | I | R | I |
| Bristol | | O | T | | | |
| Brno | | OR | | R | | |
| Bruxelles | | O | T | | R | |
| Bucuresti | | O | T | I | | I |
| Budapest | | O | T | I | R | R |
| Burgas | | | I | I | R | I |
| Bydgoszcz | | O | | R | R | P |
| Caen | | O | T | I | R | I |
| Catania | | O | T | I | | I |
| Clermont-F. | | O | T | I | R | I |
| Czestochowa | | O | IT | R | IR | |
| Dresden | | O | | I | | P |
| Dublin | IO | | T | I | R | R |
| Dusseldorf | | O | P | I | | PI |
| Edinburgh | | O | T | I | R | |
| Eindhoven | | O | T | I | T | |
| Firenze | | O | T | | R | I |
| Frankfurt | | O | | I | | |
| Freiburg | | O | T | I | | |
| Gdansk | | O | T | R | R | P |
| Geneve | | O | T | | | |
| Genova | | | T | | | P |
| Gijon | | | | | | P |
| Glasgow | | O | | I | | |
| Goteborg | IO | | T | T | | I |
| Graz | | O | T | I | R | I |
| Grenoble | | O | T | I | R | I |

| | NH3 | NMVOC | NOX | PMCO | PM25 | SO2 |
|---|---|---|---|---|---|---|
| Hamburg | | O | | I | | I |
| Hannover | | O | T | I | | P |
| Heidelberg | | O | | | | I |
| Helsinki | | O | PT | TI | R | P |
| Iasi | O | O | T | I | R | PI |
| Karlsruhe | | O | | I | | PI |
| Katowice | | O | P | I | I | PI |
| Kiel | | O | | I | | |
| Kielce | | OI | | R | R | |
| Kobenhavn | | O | T | | | |
| Koln | | | P | | | P |
| Kosice | | | I | I | | I |
| Krakow | | O | | | R | P |
| Leeds | | | | | | P |
| Lefkosia | | O | | | | |
| Leicester | | O | T | | | |
| Leipzig | | | | | | P |
| Lemesos | | | | | | P |
| Liberec | | OR | | | R | |
| Liege | | O | IT | I | R | I |
| Lille | | O | T | I | | I |
| Linz | | | | | | I |
| Lisboa | | O | | I | I | I |
| Liverpool | | O | | I | | PI |
| Ljubljana | | O | T | | R | |
| Lodz | | O | P | R | R | P |
| London | O | O | | | | |
| Lublin | | O | T | R | R | P |
| Luxembourg | | O | T | T | R | I |
| Lyon | | O | T | I | | I |
| Madrid | | O | T | | | |
| Malaga | | O | T | | | |
| Malmo | | O | T | T | | |
| Manchester | | O | T | | | |
| Mannheim | | O | I | I | | PI |
| Marseille | | | | | | I |
| Metz | | | | | | P |
| Milano | | O | T | | | |
| Modena | | O | T | I | R | I |
| Montpellier | | O | T | I | | I |
| Mulhouse | | O | T | I | R | I |
| Munchen | | O | T | I | | |
| Nancy | | | | | | I |
| Nantes | | O | T | I | | P |
| Napoli | | O | T | | R | |
| Newcastle | | | | I | | |
| Nice | | O | T | | | |
| Nottingham | | O | | | | PI |
| Nurnberg | | O | T | I | | |
| Orleans | | O | T | I | R | I |

| | NH3 | NMVOC | NOX | PMCO | PM25 | SO2 |
|---|---|---|---|---|---|---|
| Oslo | | O | | | | |
| Ostrava | | OR | PI | IO | R | PI |
| Oviedo | | | | | | P |
| Padova | | O | T | I | R | I |
| Palermo | | O | T | | R | |
| Palma de M. | | O | | | R | |
| Paris | | O | | | | |
| Parma | | O | T | I | R | I |
| Plovdiv | | | I | I | R | |
| Plzen | | O | | | R | P |
| Porto | | O | T | I | | I |
| Poznan | | O | | R | R | P |
| Praha | | OR | | | R | P |
| Rennes | | O | T | I | R | I |
| Riga | | O | | I | | |
| Roma | | O | T | | R | |
| Rotterdam | | | | | | I |
| Rouen | | O | TI | I | R | PI |
| Ruhrgebiet | | | | I | | I |
| Rzeszow | | O | | R | R | |
| Saarbrucken | | | | I | | PI |
| Santander | | | | | | PI |
| Sevilla | | O | T | | R | |
| Sheffield | | O | | I | | |
| Sofia | | O | T | I | R | P |
| Stockholm | | O | T | | | |
| Strasbourg | | O | T | I | R | I |
| Stuttgart | | O | | I | | |
| Szczecin | | O | P | | | P |
| Tallinn | | O | | I | I | P |
| Taranto | | | | | | I |
| Thessaloniki | | O | | I | I | |
| Torino | | O | T | | R | I |
| Toulouse | | O | T | I | | I |
| Tours | | O | T | I | R | |
| Usti nad Labem | | | | | | P |
| Utrecht | | O | T | | T | |
| Valencia | | O | | | | |
| Valletta | | O | P | | | P |
| Varna | I | | | | I | |
| Venezia | | | | | | P |
| Verona | | O | T | | R | I |
| Vilnius | | O | T | | R | |
| Warszawa | | O | T | R | R | P |
| West Midlands | | O | | | I | |
| Wien | | O | | I | | |
| Wroclaw | | O | | R | RI | P |
| Zagreb | | O | T | I | R | P |
| Zaragoza | | O | | | | |
| Zurich | | O | T | | | |

*Code and data availability.* The IDL code and associated input data are archived on Zenodo (https://zenodo.org/record/5654911)

*Author contributions.* PT developed the screening approach and wrote most of the text. EP prepared and formatted the input emission data and contributed to the text. AC contributed to the methodological aspects and contributed to the writing on this. JK contributed to the analysis of the results and provided text on this. BB, SLA and MG reviewed and contributed to the text.

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
