# Peer review of "A multi-pollutant and multi-sectorial approach to screen the consistency of emission inventories"

_Geoscientific Model Development, 2021_

## Author Comment (AC1)

**Review 1**

The theme discussed in the paper is of clear interest: the quality of the emission inventory (in relation to total quantity of emission, sectors of emission and spatial distribution of emissions) is of primary importance in the scenarios studies, that should be the starting points of all the plans but also of all the sustainability or integrated analysis. The quality of the emissions inventories is essential for the reliability of the studies driving policies and, so, to the effectiveness of the policies. So, the proposal methodology can be a valid instrument for validating the emission inventories.

On the other hand, it should be underlined that the methodology is not able to check the quality of an emission inventory in an absolute way but also as intercomparsion of results. If the same methodological mistake affects both the emission inventories evaluated, the method is not able to point out them. The methodology, furthermore, permits to find differences but it is not able, alone, to show what inventory is better. In any case it can help the analysis.

We agree that the method does not allow to check the quality of an inventory in an absolute way. However, the spotted inconsistencies are often so large that one (or both) inventories must be wrong. Given the magnitude of the inconsistency, it becomes very likely that the reason for this inconsistency can be found and explained, and the inconsistency corrected in most cases. This is becomes an absolute improvement. We tried to stress this point in the text at lines 172-175.

*Although very large differences may result from methodological choices as well (e.g. inclusion or not of condensable emissions for the residential sector), they are more likely to be associated to errors. Given the magnitude of the differences, it will in most cases be possible to identify one best value out of the two inventory estimates, even though the truth is unknown. These large differences therefore point to a list of potential issues for inventory compilers to check and fix where applicable, opening the way to potential improvement. In this work, these large differences are named "inconsistencies" and are intended as differences that are large enough to ensure that one of the two inventory values is tenable (i.e. justifiable) whereas the other is not.*

We included the sentence below in the conclusion to remind this important point:

*Although this screening does not allow to check the quality of the inventories in an absolute way, the magnitude of the differences is often large enough that it is possible to identify one best value out of the two inventory estimates, even though the truth is unknown*

Going to more specific questions, the idea to focus the analysis only to urban areas is another weakness of the work. For some areas ammonia is one of the key factor of the scenario analysis. It would be interesting also to develop methods in relation to temporal distribution of emissions (that can be another factor of uncertainty very important in scenario analysis, normally not faced by emission inventories).

We clarified the fact that the method can be generalized to other areas than urban, for example those related to agricultural intensive or industrial intensive emissions. In fact the method can freely be designed to address sub-areas of interest, regardless of their emission characteristics.

We agree that temporal distributions are another source of uncertainty. It is however impossible to include this aspect in the current approach. We added the sentence below in section 2 (line 106-107) to stress this point.

*Note that the method proposed here is designed with a focus on the spatial dimension. Other uncertainties related to emission inventories (e.g. speciation of VOC or PM, temporal distribution of the emissions) are not considered*

In the discussion it should be deepened also that the quality of an  emission inventory is related to the spatial (and temporal) scale of the studies undertaken: i.e. if a microscal model is used , the emission inventory should have the same scale. Here the discussion is focused only to a certain type of applications.

We added a couple of sentences to stress the fact that the method can be used, regardless of the scale of interest (from microscale to EU or global models). The selection of the areas of interest can then be adapted to the scope of the study, ranging from specific streets (microscale modelling) to urban/industrial/agriculture intensive areas (country or EU) to regions (EU or global modelling). We also added a note to stress the need of having the relevant expertise to interpret the results of the screening for a given application. The following lines have been added in the conclusions:

*Although only a particular example has been discussed here, the screening approach is general and can be used for other types of applications related to emission inventories. The approach can be applied to other inventory scales (e.g. regional or local) and can be tuned to address different sectors or areas. Intensive agriculture or industrial areas could for example be added to the urban agglomerations considered in this work. Because the type of expertise strongly depends on the type of applications, the expertise relevant to a given application is necessary to analyze the screening results and correct likely errors.*

---

## Author Comment (AC2)

**Review 2**

**Summary**

This paper provides an approach to screen / evaluate emissions inventories, with specific focus on the ability to evaluate multiple pollutants and multiple sectors across two sets of inventories. After describing the methodology, it is applied to evaluate two versions of the CAMS-REG emissions inventories over 150 cities in Europe. Given the key role of emissions inventories in air quality management, specifically as inputs in air quality models, the modeling community is always on the lookout for evaluating emissions inventories, and thus new methods for assessing / evaluating inventories are relevant and of potential interest.

**Comments:**

It is not clear how this approach differs from previous work by the same lead author published in Thunis et al, 2016. (Thunis, P., Degraeuwe B., Cuvelier C., Guevara M., Tarrason L., Clappier, 2016: A novel approach to screen and compare emission inventories. Air Qual Atmos Health 9, 325–333.)  The diamond diagram approach has been previously published in that study to screen and evaluate emissions inventories. So, the statement "we propose and discuss a screening method to compare two emission inventories" in the abstract does not seem to be justified. Further, it would have been helpful to have additional discussion in the Introduction section to see where the previous study stopped and where new improvements are made. There is no mention of this work till Line 236 when a passing reference is made as to how the Diamond diagram differs between this work and the earlier work by the authors. Further the authors state that "the diagram proposed here does not distinguishes between acceptable (within the diamond) and non-acceptable data (outside the diamond) but displays only inconsistencies". So, this seems like only an incremental tweak, and should have been presented on those lines. Suggest that the paper be rewritten and if the authors wish so, be resubmitted as a modified version of the previously published approach in Thunis et al, 2016 and present it as an application of this method for evaluating two versions of the CAMS-REG inventories.

We agree with the reviewer and added a paragraph in the introduction to stress the differences with previous works. In particular we stressed the following: while both the old and this approach share the same graphical representation, they differ in terms of concepts and scope. This is why we believe this work deserves a publication per se. While Thunis 2016 work is based on a probabilistic approach that attempts to distinguish inconsistencies in terms of "emission factor" and "activity", this approach is based on factual reported numbers that distinguish inconsistencies in terms of macro-sector shares, in terms of country totals and spatial disaggregation, and add therefore gridded data in the analysis. While the overall scope for both approaches is to detect inconsistencies, the current approach allows a straightforward investigation for the reason behind the inconsistency because input used in the screening are directly connected to the inventory input (this was not the case with emission factors and activity data). Moreover the two set of data were aggregated in terms of macro-sectors in the previous approach which further complexifies the explanation of the inconsistencies. These limitations are lifted with the current approach. We added the following two paragraphs in the introduction

*In previous works, Thunis et al. (2016) proposed a methodology to compare two emission estimates over a given area based on a limited input, the total emission per pollutant and macro sector. The differences between the total emissions of the two inventories were apportioned in terms of emission factors and activity differences. This information could then be used by emission inventory developers to identify the main causes for these discrepancies and likely errors in their estimates. However, this method is able to apportion differences between emission factors and activity only when the difference in emission factors is known for at least one of the emitted pollutants. Since this was leading to arbitrarily treat one pollutant, Clappier and Thunis (2020) improved the method by implementing a probabilistic approach to find the most likely allocation between emission factor and activity to remove this limiting assumption. In their work, Trombetti et al. (2018) then extended the approach to multiple inventories.*

*While the proposed approach shares some graphical representation with the previous, it differs in terms of diagnostics. The three original features of the new approach are: 1) differences in total emissions are allocated into three key components that provide information on the sectoral and spatial shares of the emissions at two geographical scales for each pollutant. 2) the capability to perform the analysis simultaneously for a large number of locations while systemically excluding emissions that are not relevant (lower emissions compared to others) and 3) to rank the largest inconsistencies between the two inventories.*

While this approach helps compare two inventories, a key limitation is precisely that, i.e., it is only applicable when two versions of an inventory exist. It does not provide any insight into potential biases/uncertainties in a base (or single) inventory. So, potential uncertainties that are present in both will likely be masked.

It is true that the approach can be applied only if two inventories are available, either different versions of the same inventory (as proposed here) or to two inventories that differ in terms of approach. In general inventories are regularly updated and new versions can be compared to the previous ones with the proposed approach. Obviously "errors" that are similar in both inventories will be overlooked. However, when comparing inventories that differ in their approach, it is likely that these "similar" errors will not be so frequent.

Additional specifics on the input data used in the case study will be helpful. In lines 90-101, there is reference to gridded inventory and then aggregating to city-scale or country-scale. Since spatial resolution is a key aspect of the emissions inventories, the spatial resolutions of interest and being used are not clear. Are the authors going from grid-scales to city-scales or country-scales? What about potential issues about non-urban areas where some sectors (e.g., agricultural) may be of more importance?

We clarified the aspects related to the spatial resolution. We also clarified the fact that the method can be generalized to other areas than urban, for example those related to agricultural or industrial emissions. In fact the method can freely be designed to address sub-areas of interest, regardless of their emission characteristics.

It is not clear as to how the authors chose to present inconsistencies for the 4 examples (3 in the UK and one in Lithuania) discussed in detail. Agreed these are illustrative examples but

additional context as to how/these were picked can be helpful.  Can the authors also provide additional context as to how important these 4 cities are compared to the 150 cities studied in terms of emissions magnitudes? Also, the ECI metric bins these 4 examples into two approximate bins (two of them with 2 and 2.5, and the other two with 50 and 68). Can the authors explain this ECI pattern, and how this affects the choice of these illustrations?

The examples have been selected to illustrate different type of issues. They were however picked among those showing important inconsistencies. Depending on the pollutant and sector, the ECI can differ quite strongly explaining the two bins of ECI in our 4 examples. We added the following sentences in the text to clarify this.

*Examples are picked among those showing important inconsistencies with the aim to illustrate different type of inconsistencies, i.e. LSS, FAS and LPT (**Error! Reference source not found.**). Note that depending on pollutant and sector, the ECI can differ quite strongly in magnitude explaining the different range of values in our examples.*

A key gap in the discussion is how the outputs of this approach in terms of emissions inconsistencies can be used further. Given the detailed discussion of the methods and the illustrative examples, the paper can be strengthened by how this can be put to practical uses. There is some discussion in the Conclusions section  (Lines 490-492) that half of the inconsistencies in the studies example is due to differences in country-wise reporting. Given that, wouldn't it be expected that this can be potentially easily addressed by promoting a harmonized approach among EU countries for development of emissions inventories?

In the specific case addressed in this work, the method has been used to correct for some of the spotted inconsistencies in terms of spatial distribution. This was possible as these issues were directly under control of the emission developer teams. This is unfortunately not the case for inconsistencies related to country totals where information is not always available. In the example of the CAMS-REG inventory, if the likely error is in the earlier version but not in the newer version anymore, it may also mean that it was already found and corrected by the inventory team. Note that the method is not able to spot a lack of harmonization across countries but will only identify an inconsistency between two reporting for a given country

**Minor Comments:**

- An easy to access list of the 150 cities, 4500 screened pollutant sectors, 450 relevant ones, and the 46 that showed inconsistencies would be helpful. Further, some quantitative information on these 46 vs 450 vs 4500 would add even more value to the paper.

  We added an annex that distinguishes for each city the non-relevant and relevant emissions and among the latter those that lead to inconsistencies, for each sector and pollutant.

- In Figure 4, can you clarify how the 46 points map to a subset of the 150 cities that were studied?

A mapping based only on that figure is not possible. It can partially be done in combination with Figure 5 and fully with the application itself from which output similar to figure 4 can be produced city by city. Now, the new annex facilitates this connection.

- Lines 65-66: Add "etc." at the end of the sentence

Done

- Figure 6: Can this be made larger, and use darker / bolder font? This figure is mostly difficult to read

We improved the layout of Figure 6

- References: Thunis et al, 2013 and Thunis et al, 2016 seem repeated, but with inconsistent years

Indeed, this has been corrected

**Citation**: https://doi.org/10.5194/gmd-2021-390-RC2

---

## Author Response (AR2)

**Comments to the author:**

Please consider updating the sentence "As only air quality models can simulate the impacts of emission reductions considering the complexity of the atmosphere, they are potentially the only tools able to support the planning of reduction strategies" to reflect that a multitude of data sources (e.g. ambient observations) and tools are often used to plan reduction strategies.

We have changed the beginning of the sentence to "As only models can simulate the impacts…". This is to account for other type of models than air quality models, like receptor models. We however did not add observations explicitly because although they can be used to improve or calibrate the model baseline, observations cannot be used directly to simulate the impact of emission reductions.

**Non-public comments to the Author:**

**Reviewer #2's most significant concern was the possible duplication of this work with a previous 2016 manuscript. After reviewing the 2016 paper, there seemed to be enough new content to avoid duplication.**

**Figure 6 may need some formatting as each panel is different in size.**

We reformatted the figure

**Notification to the authors**:

1. For the next revision, please rename the Table 1 in section "6. Appendix" to "Table A1". See more: https://www.geoscientific-model-development.net/submission.html#manuscriptcomposition / 7. Appendices /. 2. Your tables contain coloured cells or/and coloured values. Please note that this will not be possible in the final revised version of the paper due to HTML conversion of the paper. When revising the final version, you can use footnotes or italic/bold font. But if the colour spectrum is necessary and cannot be exchanged for footnotes, bold, or italic, then please inform us via email. 2. Please ensure that the colour schemes used in your maps and charts allow readers with colour vision deficiencies to correctly interpret your findings. Please check your figures using the Coblis – Color Blindness Simulator (https://www.color-blindness.com/coblis-color-blindness-simulator/) and revise the colour schemes accordingly.

Updates have been done in the document.